# Effect of Fixed Orthodontic Treatment on Salivary Nickel and Chromium Levels: A Systematic Review and Meta-Analysis of Observational Studies

**DOI:** 10.3390/dj7010021

**Published:** 2019-03-01

**Authors:** Mohammad Moslem Imani, Hamid Reza Mozaffari, Mazaher Ramezani, Masoud Sadeghi

**Affiliations:** 1Department of Orthodontics, Kermanshah University of Medical Sciences, Kermanshah 6713954658, Iran; m.imani@kums.ac.ir; 2Department of Oral and Maxillofacial Medicine, School of Dentistry, Kermanshah University of Medical Sciences, Kermanshah 6713954658, Iran; mozaffari@kums.ac.ir; 3Medical Biology Research Center, Kermanshah University of Medical Sciences, Kermanshah 6714415185, Iran; 4Molecular Pathology Research Center, Imam Reza Hospital, Kermanshah University of Medical Sciences, Kermanshah 6714415153, Iran; m_ramezani@kums.ac.ir; 5Students Research Committee, Kermanshah University of Medical Sciences, Kermanshah 6715847141, Iran

**Keywords:** Orthodontic appliances, saliva, nickel, chromium, meta-analysis

## Abstract

Nickel and chromium ions released from fixed orthodontic appliances may act as allergens. This study aimed to systematically review the effect of fixed orthodontic treatment on salivary levels of these ions by doing a meta-analysis on cross-sectional and cohort studies. The Web of Science, Scopus, Cochrane Library, and PubMed databases were searched for articles on salivary profile of nickel or chromium in patients under fixed orthodontic treatment published from January 1983 to October 2017. A random-effect meta-analysis was done using Review Manager 5.3 to calculate mean difference (MD) and 95% confidence interval (CI), and the quality of questionnaire was evaluated by the Newcastle–Ottawa scale. Fourteen studies were included and analyzed in this meta-analysis. Salivary nickel level was higher in periods of 10 min or less (MD = −11.5 µg/L, 95% CI = −16.92 to −6.07; *P* < 0.0001) and one day (MD = −1.38 µg/L, 95% CI = −1.97 to −0.80; *P* < 0.00001) after initiation of treatment compared to baseline (before the insertion of appliance). Salivary chromium level was higher in periods of one day (MD = −6.25 µg/L, 95% CI = −12.00 to −0.49; *P* = 0.03) and one week (MD = −2.07 µg/L, 95% CI = −3.88 to −0.26; *P* = 0.03) after the initiation of treatment compared to baseline. Corrosion of fixed orthodontic appliances leads to elevated salivary nickel and chromium concentrations early after initiation of orthodontic treatment. Randomized clinical trials controlling for factors affecting the saliva composition are recommended on a higher number of patients and among different ethnicities.

## 1. Introduction

Orthodontic appliances are highly biocompatible, although some side effects associated with the release of nickel ions have been documented [1]. Fixed orthodontic appliances including brackets and arches are commonly made of stainless steel and nickel–titanium (NiTi) alloys and, therefore, have corrosion potential in the oral environment [2]. The amount of nickel as the main constituent of contemporary orthodontic appliances [3] may vary from 8% in stainless steel [4,5] to more than 50% in NiTi alloys [4]. Stainless-steel alloys include 17% to 22% of chromium [6]. Fixed orthodontic treatment causes major changes in the composition of the saliva [7]. Nickel and chromium ions released from fixed orthodontic appliances can serve as allergens or may have serious biological side effects [2,6]. Moreover, they are cytotoxic, mutagenic, and carcinogenic in small quantities in the range of nanograms [2]. Evaluation of the level of trace elements in patients using orthodontic appliances is a priority [8]. Both nickel and chromium ions can cause hypersensitivity reactions in some people [9]. In addition, nickel and chromium can cause dermatitis and asthma [10]. Increased prevalence of nickel hypersensitivity as well as the increased demand and availability of orthodontic treatment have attracted the attention of researchers towards the composition of alloys and their ion release potential during orthodontic treatment [9]. Orthodontic appliances (brackets and wires) exposed to the oral environment are affected by thermal alterations in the oral cavity and pH, constant presence of saliva, exposure to foods and drinks, mechanical loads applied to them, and abrasion. They are subjected to aging as such and may undergo dissolution or oxidation [1,11,12]. The placement of archwires can cause an increase in salivary nickel and chromium levels and, therefore, nickel may be released from the wires [7,13] as well as bands and brackets [13]. Daily oral intake of nickel from food is estimated to be 200–300 to 600 μg/day [1,14]. The average dietary intake of chromium is estimated to be 50–200 μg/day [15]. In vitro nickel release from orthodontic appliances was reported to be 22–40 μg/day, which was lower than the estimated dietary intake [5]. The inherent heterogeneity of metal alloys and their use in combination with other alloys, microconversion, the forces acting on the appliances, and the friction between wires and brackets may further add to the corrosion process [16]. Therefore, in orthodontic practice, it is essential to know the exact amount of each ion released during the course of treatment, and inform the patient undergoing orthodontic treatment in this respect [2].

Controversy exists regarding the reported values of nickel and chromium ions released during fixed orthodontic treatment in previous studies; therefore, we decided to systematically review the previous studies on the effect of fixed orthodontic treatment on salivary levels of nickel and chromium ions in a meta-analysis of observational studies. 

## 2. Materials and Methods

This meta-analysis was done according to the guidelines for the preferred reporting items for systematic review and meta-analysis (PRISMA) [17].

### 2.1. Search Strategy 

We searched the Web of Science, Scopus, Cochrane Library, and PubMed databases for articles on assessment of salivary profiles of nickel or chromium in patients under orthodontic treatment using the search terms "nickel or chromium", "orthodontic", and "saliva or salivary". The search was limited to human studies accessible in PubMed, Web of Science, and Scopus published from January 1983 to October 2017.

### 2.2. Study Selection

Articles on evaluation of salivary profile of nickel or chromium in patients under fixed orthodontic treatment (Table 1) were selected after reviewing their English abstract. Studies were included if they had a cross-sectional, cohort, or case-control design and evaluated salivary profile, including the mean/median salivary nickel or chromium level. Review articles, letters to editor or case reports, and articles with full texts that were not accessible free of charge or that had not reported the mean or median salivary concentration of ions were excluded. The most important inclusion criteria were as follows: 1) patients with no previous history of orthodontic treatment, 2) presence of controls without previous/current orthodontic treatment, 3) patients and controls without any systemic disease or dental problem, and 4) saliva samples collected from patients and controls after rinsing the mouth.

### 2.3. Data Extraction from Studies

One author (M.S.) reviewed the articles, screened the titles and abstracts based on the aforementioned criteria, and extracted the required data. Two authors (M.R. & M.S.) independently rechecked the full text of the screened studies. Data collected from each study included the name of the first author, year of publication, country, number of patients evaluated, percentage of male patients, mean age of patients, and salivary nickel or chromium levels. These data were rechecked by another author (M.M.I.). 

### 2.4. Quality Evaluation

The quality of the questionnaire was evaluated using the Newcastle–Ottawa scale [18], in which the maximum total score is 10 for a cross-sectional study and 9 for a cohort or case-control study. The quality evaluation was done by two authors (M.R. & M.S.) for each study until a consensus was reached via discussion.

### 2.5. Statistical Analyses

A random-effect meta-analysis was done by the Review Manager 5.3 (The Cochrane Collaboration, Oxford, United Kingdom) using the mean difference (MD) and 95% confidence interval (CI). Heterogeneity between the estimates was evaluated by the Q and I2 statistic. For the Q statistic, heterogeneity was considered if *P* < 0.1, and *P*-value (2-sided) <0.05 was considered statistically significant in this meta-analysis. The heterogeneity (I^2^) statistic yields results ranging from zero to 100%; 0 to 25% indicates lack of heterogeneity, 25% to 50% indicates moderate heterogeneity, 50% to 75% indicates large heterogeneity, and 75% to 100% indicates extreme heterogeneity [19]. In addition, the publication bias was evaluated by the funnel plot analysis and the Begg’s and Egger’s tests. We used a formula for estimation of the mean and standard deviation if the study reported median plus range [20], or median plus interquartile range [21]. The unit of measurement of salivary nickel or chromium level was microgram per liter (µg/L). 

## 3. Results

### 3.1. Characteristics of Included Studies

The search of the four databases yielded 451 studies (Figure 1). After excluding the duplicates, 283 studies were screened, from which 262 were not relevant according to the set criteria. Therefore, the full texts of 21 studies were assessed for eligibility. After reading the full texts of 21 studies, 7 studies were excluded with reasons (two studies had invalid data, one case-control study had lost data, one case-control study had reported the level of ions in the mass of saliva instead of volume, two randomized clinical trials had heterogeneous methods, and one study had different number of patients in each treatment period). Finally, 14 articles were included and analyzed in this meta-analysis.

Table 2 shows the characteristics of 14 studies included in this meta-analysis. Four studies had been conducted in India [2,10,22,23], three in Iran [24,25,26], one in Norway [5], one in Turkey [13], one in Brazil [11], one in Greece [4], one in Morocco [6], one in Saudi Arabia [15], and one in Germany [1]. Two studies [1,22] reported the median (quartile) of the salivary level of ions, two studies [4,5] reported the median (range), and one study [6] reported the mean (range) values. Three studies [2,6,23] had only evaluated one assessment period of orthodontic treatment. Three studies did not report the mean age [1,2,11]. Table 2 shows the mean age, age range, percentage of male and female patients, follow-up periods, analysis methods, and saliva sampling method in each study.

### 3.2. Salivary Nickel and Chromium Levels (before versus after Fixed Orthodontic Treatment)

#### 3.2.1. Nickel Levels in 10 Time Periods

Figure 2 shows the salivary nickel levels in 10 time periods before and after the insertion of fixed appliances. The pooled MD was estimated for the periods of 10 min or less (MD = −11.5 µg/L, 95% CI = −16.92 to −6.07; *P* < 0.0001), 1 day (MD = −1.38 µg/L, 95% CI = −1.97 to −0.80; *P* < 0.00001), 1 week (MD = 1.02 µg/L, 95% CI = −2.16 to 4.21; *P* = 0.53), 3 weeks (MD = −0.81 µg/L, 95% CI = −3.04 to 1.43; *P* = 0.48), 1 month (MD = 5.53 µg/L, 95% CI = −1.42 to 12.48; *P* = 0.12), 2 months (MD = 3.42 µg/L, 95% CI = 0.83 to 6.00; *P* = 0.010), 3 months (MD = 2.63 µg/L, 95% CI = 0.77 to 4.49; *P* = 0.006), 6 months (MD = −0.03 µg/L, 95% CI = −1.87 to 1.80; *P* = 0.97), 12 months (MD = −0.66 µg/L, 95% CI = −4.15 to 2.83; *P* = 0.71), and 24 months (MD = 0.01 µg/L, 95% CI = −0.95 to 0.97; *P* = 0.98) after fixed orthodontic treatment compared to baseline (before fixed orthodontic treatment). There was no heterogeneity in 10 min or less, 1 day, 3 weeks, 2 months, and si6x months, whereas extreme heterogeneity was found in 1 week and 1 month, and a large heterogeneity was observed in 12 months. There were significant differences in time periods of 10 min or less, 1 day, 2 months, and 3 months. In time periods of 10 min or less and 1 day, salivary nickel level was higher after treatment compared to before treatment, but in periods of 2 months and 3 months, salivary nickel level after treatment was less than that before treatment. 

#### 3.2.2. Chromium Levels in 10 Time Periods

Figure 3 shows the salivary chromium levels in 10 time periods after the insertion of fixed appliances. The pooled MD was estimated for the time periods of 10 min or less (MD = −12.51 µg/L, 95% CI = −42.47 to −6.07; *P* = 0.42), 1 day (MD = −6.25 µg/L, 95% CI = −12.00 to −0.49; *P* = 0.03), 1 week (MD = −2.07 µg/L, 95% CI = −3.88 to −0.26; *P* = 0.03), 3 weeks (MD = −1.10 µg/L, 95% CI = −2.84 to 0.65; *P* = 0.22), 1 month (MD = 1.16 µg/L, 95% CI = −2.68 to 5.00; *P* = 0.56), 2 months (MD = 0.12 µg/L, 95% CI = −0.16 to 0.40; *P* = 0.41), 3 months (MD = −0.81 µg/L, 95% CI = −2.40 to 0.78; *P* = 0.32), 6 months (MD = −0.78 µg/L, 95% CI = −2.16 to 0.61; *P* = 0.27), 12 months (MD = 0.72 µg/L, 95% CI = −1.21 to 2.65; *P* = 0.47), and 24 months (MD = 0.22 µg/L, 95% CI = −0.09 to 0.53; *P* = 0.16) after insertion compared to baseline (before fixed orthodontic treatment). There was no heterogeneity in 3 weeks and 6 months after treatment, whereas an extreme heterogeneity was found in 1 day, 1 week, and 12 months and a large heterogeneity in 10 min or less and 1 month after treatment. There were significant differences in time periods of 1 day and 1 week; salivary chromium level was found to be higher after treatment than before treatment in both periods.

#### 3.2.3. Nickel Levels in Five Time Periods

Figure 4 shows salivary nickel levels in five time periods after the insertion of fixed appliances. The pooled MD was estimated immediately after placement of the bands and brackets (MD = −36.79 µg/L, 95% CI = −56.08 to −17.51; *P* = 0.0002), after two weeks (MD = 8.27 µg/L, 95% CI = 3.75 to 12.78; *P* = 0.0003), before placement of archwires (MD = −20.27 µg/L, 95% CI = −38.24 to −2.31; *P* = 0.03), immediately after placement of archwires (MD = 12.46 µg/L, 95% CI = 8.45 to 16.46; *P* < 00001), four weeks after placement of wires (MD = 8.14 µg/L, 95% CI = 4.66 to 11.63; *P* < 00001), and eight weeks after placement of wires compared to baseline (before fixed orthodontic treatment). There was a large heterogeneity in the first and third periods, whereas there was no heterogeneity in the second, fourth, and fifth periods. There were significant differences in all periods such that the salivary nickel level was higher after placement of the bands and brackets and immediately after placement of archwires compared to baseline (before treatment), but this value was lower after two weeks, before placing the archwires, four weeks after placement of wires, and four weeks after placement of wires compared to before treatment.

### 3.3. Salivary Nickel and Chromium Levels (Case-Control Studies)

#### 3.3.1. Nickel Levels 

Three studies showed MD of the salivary nickel in patients under fixed orthodontic treatment compared to controls (Figure 5). The pooled MD was estimated at 3.14 µg/L (95% CI = −0.92 to 7.21; *P* = 0.13) in patients compared to controls. There was a moderate heterogeneity.

#### 3.3.2. Chromium Levels

Figure 6 shows MD of the salivary chromium level in patients compared to controls in three studies. The pooled MD was estimated at −0.09 µg/L (95% CI = −0.85 to 0.67; *P* = 0.82) in patients compared to controls. There was a large heterogeneity.

#### 3.3.3. Quality Evaluation

Table 3 shows the quality score for each study included in the meta-analysis. The mean score was 7.78 for cross-sectional, 6.67 for cohort, and 7.50 for case-control studies.

### 3.4. Publication Bias

Figure 7A shows the evaluation of publication bias among the included studies on salivary nickel levels. Begg’s and Egger’s tests did not reveal significant evidence of publication bias among the included studies on salivary nickel levels in time periods of 10 min or less and after 1 month. Begg’s and Egger’s tests revealed significant evidence of publication bias in periods of after one day and after one week. Regarding the periods of after 3 weeks, after 2 months, after 3 months, after 6 months, and after 24 months, Begg’s test revealed no publication bias, and Egger’s test could not be performed because only two studies were included. For the period of after 12 months, Begg’s test revealed no publication bias, but Egger’s test revealed a significant publication bias. Figure 7B shows evaluation of the publication bias among the included studies on salivary chromium level. Begg’s and Egger’s tests did not reveal significant evidence of publication bias among the included studies on salivary chromium levels in periods of after 1 month and after 12 months. Begg’s and Egger’s tests revealed significant evidence of publication bias among the included studies on salivary nickel level in time periods of after one day and after one week. As for the period of 10 min or less, after 3 weeks, after 2 months, after 3 months, after 6 months, and after 24 months, Begg’s test revealed no publication bias, and Egger’s test could not be performed because only two studies were included.

## 4. Discussion

This meta-analysis evaluated the salivary nickel and chromium levels in patients under fixed orthodontic treatment for different periods of time compared to controls. The salivary nickel level was higher in periods of 10 min or less and 1 day from the insertion of appliance as well as after placement of the bands and brackets and immediately after placement of archwires compared to baseline, while the salivary nickel level was lower than the baseline (pretreatment) value in periods of after two months and three months from the insertion of appliance as well as two weeks after the insertion of appliance except for wires, and four weeks after the placement of wires. The salivary chromium level was higher than the baseline pretreatment value in periods of one day and one week after the insertion of appliance. In this meta-analysis, the mean concentration of chromium in the reviewed studies ranged from 0.68 to 142.16 µg/L and the mean concentration of nickel ranged from 0.52 to 102.68 µg/L.

It has been shown that nickel and chromium ions can cause hypersensitivity reactions, dermatitis, and asthma; thus, nickel and chromium ions released from stainless-steel orthodontic bands, brackets, and wires are likely to cause allergic reactions [5,27]. The composition of saliva may be affected by many physiological variables such as time of the day, health conditions, diet [28], and salivary flow rate [29]. The emotional state also affects the salivary flow rate; for example, anxiety and depression can cause dry mouth [30]. The large variations in nickel levels reported in studies might be explained by the differences in saliva composition and pH, which are influenced by various physiological and environmental factors such as time of the day, diet, health, and mental conditions as well as nickel adhesion to epithelial cells, bacteria, macromolecules of the saliva [9,11,31,32,33], and the method of sampling (stimulated versus unstimulated saliva collection) [7,19,34]. Differences in the methodology, sample size, and time periods of sample collection may explain generally lower nickel concentrations in patients treated with fixed appliances [10]. High metal ion levels are found one to two weeks after exposure to metal appliances, which will later return to their initial levels [4,5,13,35]. Orthodontic appliances release ions into the oral environment [1,11,12] more intensely in the first months of placement of the appliance [1,10]. In the present study, in the periods of 10 min or less and 1 day after fixed orthodontic treatment, nickel levels significantly increased compared to baseline. In addition, chromium levels significantly increased at one day and one week after the initiation of treatment compared to baseline. Therefore, a passivation layer is required to reduce the release of ions. In addition, reduction in salivary level of nickel after two and three months of treatment with a peak at one month shows that the maximum release of nickel probably occurs earlier than one month but the environmental and psychological conditions of the patients undergoing treatment immediately after placement of bands, brackets, and archwires may affect the release of nickel; these conditions may be variable in different studies. 

An in vitro study demonstrated the corrosion potential of orthodontic appliances [2]. Some studies showed nickel [36,37,38,39,40] and chromium [11,37,39,41] release from orthodontic brackets. However, when placed in the oral cavity; the appliances are mechanically activated to facilitate tooth movement. The archwire movement and bracket friction may lead to greater corrosion and increase the release of metal ions from the orthodontic appliances [2]. Corrosion of orthodontic alloys may lead to the release of sizeable amounts of nickel and chromium into the saliva [42,43]. In the present study, salivary nickel levels significantly increased in two time periods, namely after placement of the bands and brackets and immediately after placement of archwires. Therefore, corrosion can be an effective factor to increase salivary nickel levels in fixed orthodontic treatment. Among the six case-control studies included in this meta-analysis, two studies [15,26] showed significantly higher nickel level in patients under orthodontic treatment than controls and two studies [32,44] did not show any significant difference between the two groups, but two other studies [9,34] showed a lower level in patients compared to controls (*P* > 0.05). Also, evaluation of chromium level in five studies revealed that three studies [26,34,44] reported a higher level in patients compared to controls (*P* > 0.05) while two other studies [9,15] reported a lower level in patients compared to controls. Meta-analysis of two eligible case-control studies [15,26] showed a higher level of nickel, but a lower level of chromium without a significant difference between the two groups of patients and controls. The difference in salivary nickel and chromium levels among in vivo or in vitro studies and also between in vivo and in vitro studies can be due to the difference in ions released from orthodontic appliances. Based on the conclusions of reviewed studies, type of appliances (bands, brackets, and archwires) having the different amounts of ions in their structure, history of allergy in patient, period of treatment, saliva pH, and diet were effective factors in the level of ions. The lack or presence of heterogeneity in time periods cannot have a reasonable explanation due to the small number of participants or studies in some time periods. Therefore, in order to have a strong clinical result, more homogeneous studies having the similar times should be undertaken. The small number of case-control studies included in this meta-analysis can affect the results as a bias. Further studies and clinical trials are required in this respect. Moreover, future studies should pay close attention to the method of saliva sampling and the follow-up time to minimize bias.

Limitations of this meta-analysis included different protocols of saliva sampling in different studies, different analyses of the saliva samples in different studies, different percentages of males and females in the studies, different age range of patients, different time periods of evaluation, and small number of participants in the majority of studies.

## 5. Conclusions

Overall, within the limitations of this meta-analysis, it can be concluded that corrosion of fixed orthodontic appliances leads to elevated salivary nickel and chromium levels early after the initiation of orthodontic treatment. The small amount of nickel released from orthodontic appliances in cross-sectional/cohort and case-control studies may support the opinion of an orally-induced tolerance level against nickel early after the initiation of orthodontic treatment. Randomized clinical trials controlling for the factors affecting the composition of saliva are recommended with a larger sample size and among different ethnic groups.

## Figures and Tables

**Figure 1 dentistry-07-00021-f001:**
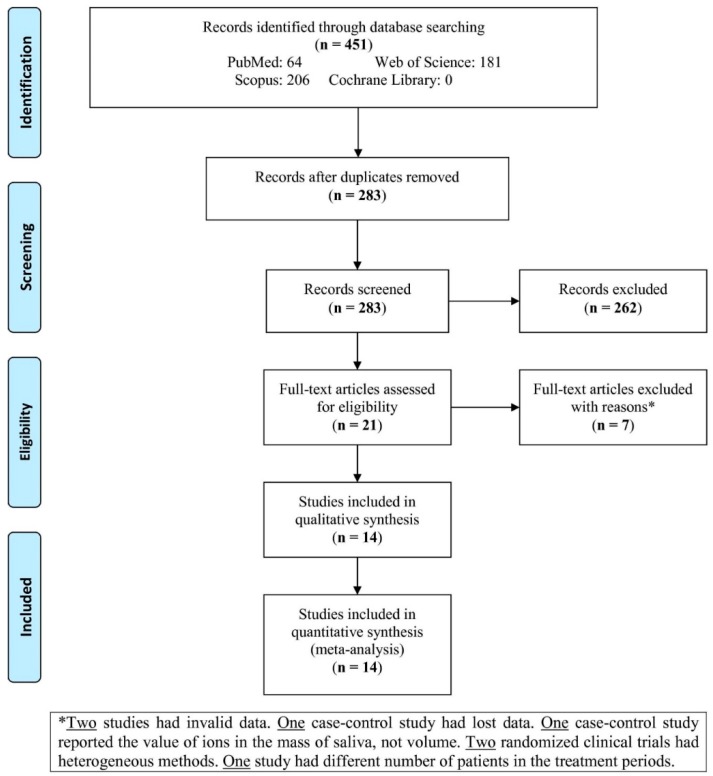
Flowchart of literature search and study selection.

**Figure 2 dentistry-07-00021-f002:**
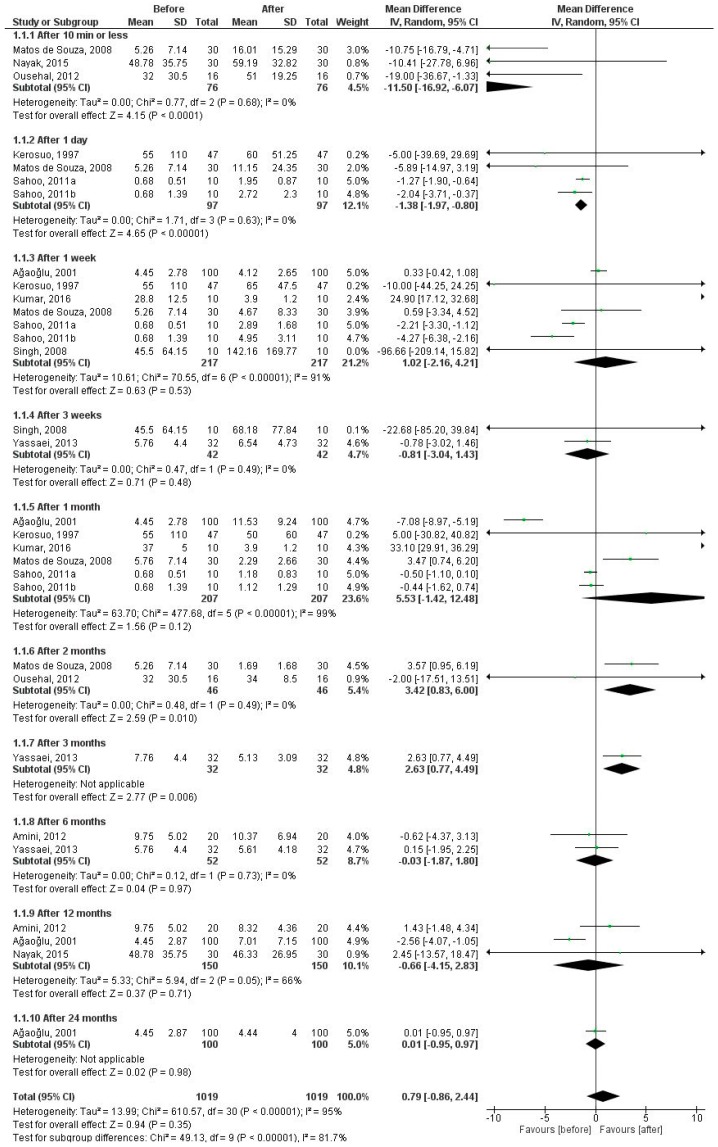
Forest plot of the random effect of salivary nickel levels in patients under fixed orthodontic treatment (before versus after treatment).

**Figure 3 dentistry-07-00021-f003:**
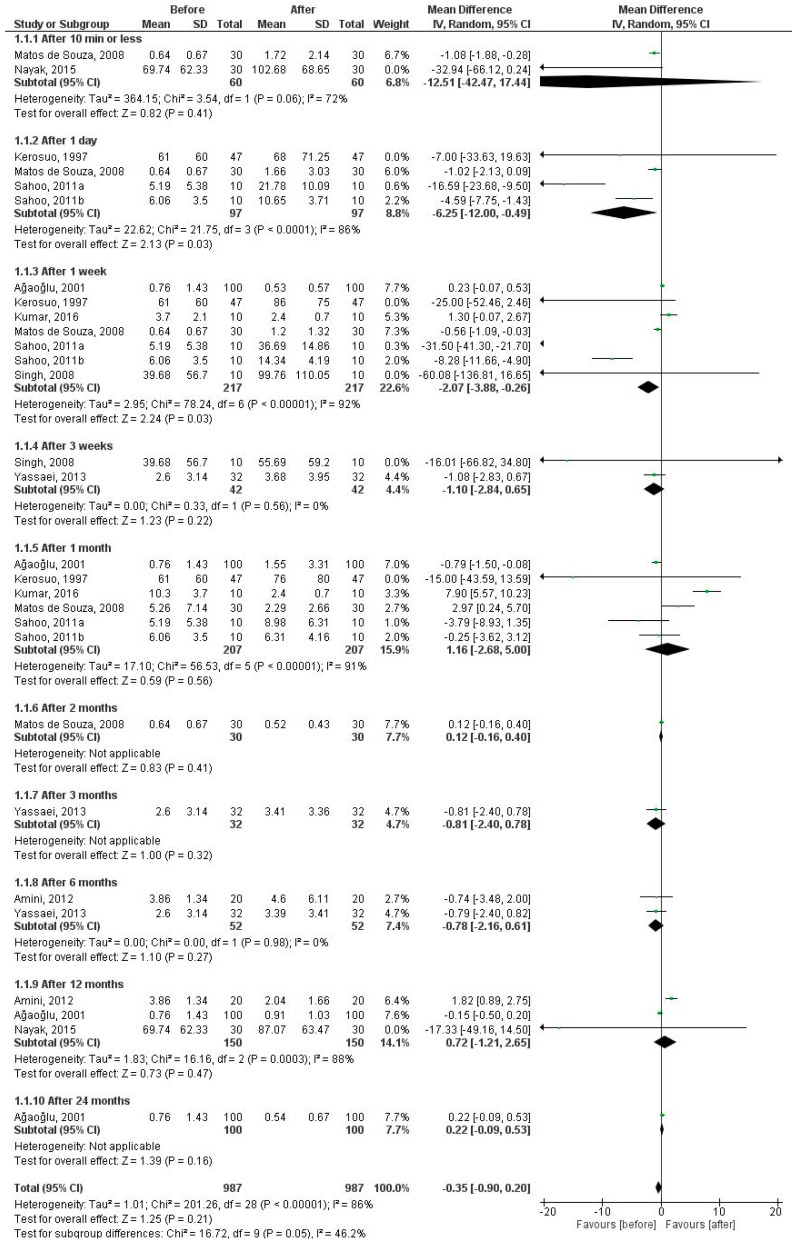
Forest plot of the random effect of salivary chromium levels in patients under fixed orthodontic treatment (before versus after treatment).

**Figure 4 dentistry-07-00021-f004:**
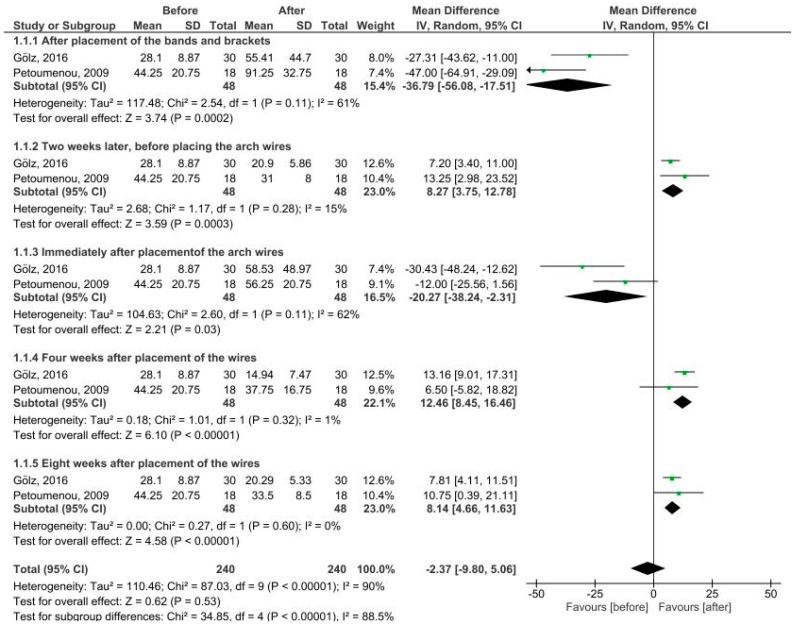
Forest plot of the random effect of salivary nickel levels in patients under fixed orthodontic treatment (before versus after treatment).

**Figure 5 dentistry-07-00021-f005:**
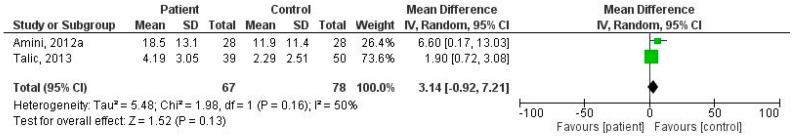
Forest plot of the random effect of salivary nickel levels in patients under fixed orthodontic treatment in case-control studies.

**Figure 6 dentistry-07-00021-f006:**
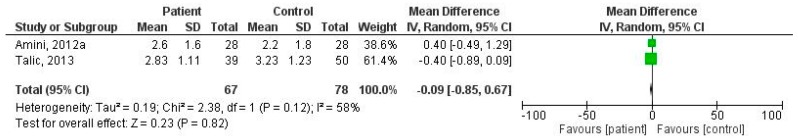
Forest plot of the random effect of salivary chromium levels in patients under fixed orthodontic treatment in case-control studies.

**Figure 7 dentistry-07-00021-f007:**
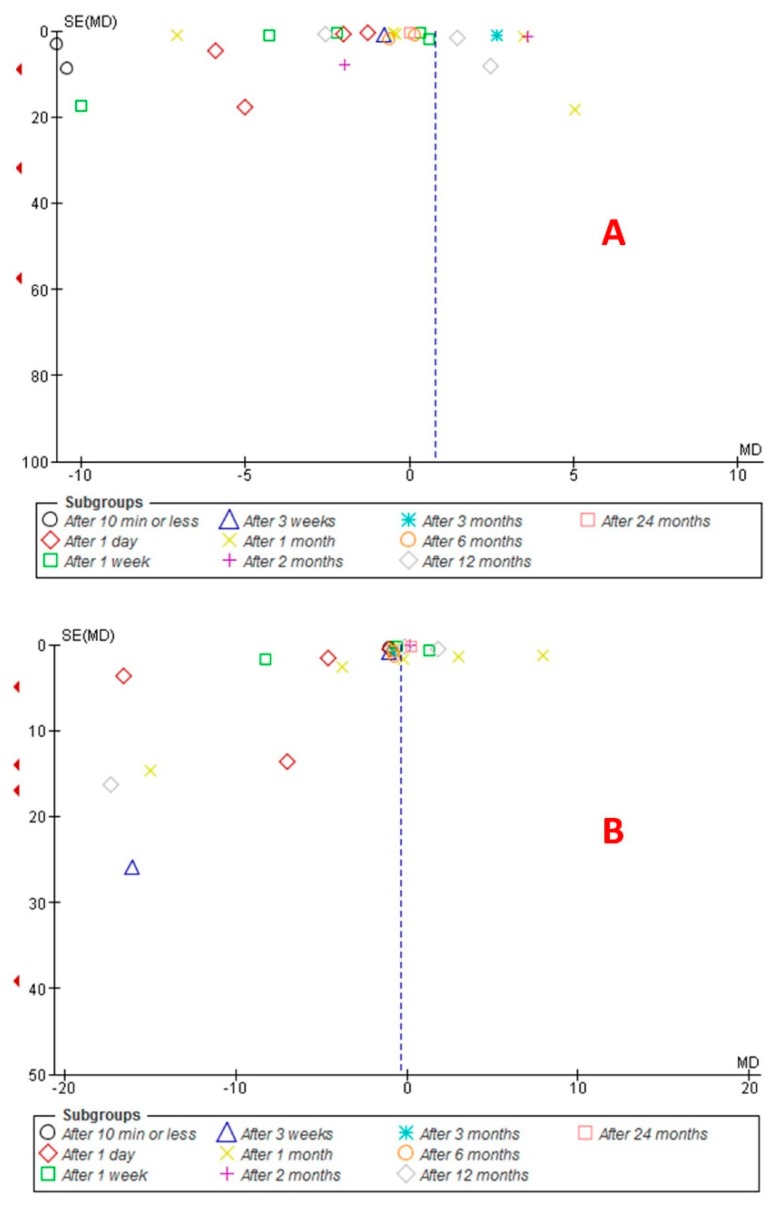
Funnel plot of the random effect of (A) salivary nickel levels and (B) salivary chromium levels in patients under fixed orthodontic treatment (before versus after treatment).

**Table 1 dentistry-07-00021-t001:** Population, intervention/exposure, comparison, and outcome of the included studies.

Criteria	Description
Population	Eligible patients for fixed orthodontic treatment
Intervention/exposure	Fixed orthodontic treatment
Comparison	Before vs after intervention/patients vs controls
Outcome	Salivary nickel and/or chromium levels

**Table 2 dentistry-07-00021-t002:** Characteristics of the studies included in this meta-analysis (n = 14).

The First Author, Year	Country	Type of Study	Participants (n)	Mean Age/Age Range of Patients, Year	Percentage of Male Patients	Outcome	Follow-Up	Analysis Method	Saliva Sampling
Kerosuo, 1997 [5]	Norway	Cross-sectional	47	12.4/8–30	40.4	Salivary nickel & chromium	1 day, 1 week, 1 month,	Atomic absorption spectrophotometry	NA/stimulated
Ağaoğlu, 2001 [13]	Turkey	Cross-sectional	100	19.5/12–33	33	Salivary nickel & chromium	1 week, 1 month, 12 months, 24 months	Atomic absorption spectrophotometry	Fasting/unstimulated
Matos de Souza, 2008 [11]	Brazil	Cross-sectional	30	-/20–26	26.7	Salivary nickel & chromium	10 min and less, 1 day, 1 week, 1 month, 2 months	Atomic absorption thermal electric spectrophotometry	Fasting/NA
Singh, 2008 [22]	India	Cross-sectional	10	17.5/14–24	30	Salivary nickel & chromium	1 week, 3 weeks,	Atomic absorption thermal electricspectrophotometer	NA/stimulated
Petoumenou, 2009 [4]	Greece	Cross-sectional	18	14.9/12–18.1	44.5	Salivary nickel	Up to 8 weeks after the placement of the wires	Atomic absorption spectrometry	Nonfasting/unstimulated
Sahoo, 2011 [10]	India	Cross-sectional	20	21.5/18–25	0	Salivary nickel & chromium	1 day, 1 week, 1 month	Atomic absorption spectrometry	NA/unstimulated
Amini, 2012 [24]	Iran	Prospective cohort	20	16/14–23	40	Salivary nickel & chromium	6 months, 12 months	Atomic absorption spectrophotometry	Fasting /unstimulated
*Amini, 2012 [26]	Iran	Case-control	28 vs. 28	17.5/16–19 vs. 18.2/14–22	42.8 vs. 42.8	Salivary nickel & chromium	12–18 months	Atomic absorption spectrophotometry	NA/NA
Ousehal, 2012 [6]	Morocco	Cross-sectional	16	~19.1/13–25	50	Salivary nickel	10 min and less, 2 months	Inductively coupled plasma–mass spectrometry	NA/unstimulated
* Talic, 2013 [15]	Saudi Arabia	Case-control (cross-sectional)	39 vs. 50	18.1 vs. 22	41 vs. 48	Salivary nickel & chromium	1–32 months	Inductively coupled plasma–mass spectrometry	NA/unstimulated
Yassaei, 2013 [25]	Iran	Cohort	32	~15.3/11–24	NA	Salivary nickel & chromium	3 weeks, 3 months, 6 months	Atomic absorption spectrometry	NA/unstimulated
Nayak, 2015 [2]	India	Cross-sectional	30	-/10–25	50	Salivary nickel & chromium	10 min and less, 12 months	Inductively coupled plasma–mass spectrometry	NA/NA
Kumar, 2016 [23]	India	Cross-sectional	10	NA/14–23	50	Salivary nickel & chromium	10 days, 1 month	Inductively coupled plasma–optical emission spectrometry	NA/NA
Gölz, 2016 [1]	Germany	Prospective cohort	30	-/10–13	NA	Salivary nickel	Up to 8 weeks after the placement of the wires	Inductively coupled plasma–mass spectrometry	Nonfasting/unstimulated

* Data in the study = patients vs controls

**Table 3 dentistry-07-00021-t003:** Quality ratings for the studies included on the basis of Newcastle–Ottawa quality assessment scale (n = 14).

First Author, Publication Year	Selection	Comparability	Outcome	Total Score
Kerosuo, 1997 [5]	4	1	3	8
Ağaoğlu, 2001 [13]	4	2	3	9
Matos de Souza, 2008 [11]	3	1	3	7
Singh, 2008 [22]	3	1	3	7
Petoumenou, 2009 [4]	4	2	3	9
Sahoo, 2011 [10]	3	2	3	8
Ousehal, 201 [6]	3	2	3	8
Nayak, 2015 [2]	3	0	3	6
Kumar, 2016 [23]	3	2	3	8
Mean score (cross-sectional studies)				7.78
Amini, 2012 [24]	2	2	3	7
Yassaei, 2013 [25]	2	2	3	7
Gölz, 2016 [1]	2	2	2	6
Mean score (cohort studies)				6.67
Amini, 2012 [26]	4	2	2	8
Talic, 2013 [15]	3	2	2	7
Mean score (case-control studies)				7.50

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
