# Peer review of "Effect of Fixed Orthodontic Treatment on Salivary Nickel and Chromium Levels: A Systematic Review and Meta-Analysis of Observational Studies"

_dentistry, 2019, doi:10.3390/dj7010021_

Round 1

Reviewer 1 Report

Reviewdentistry journal

dentistry-419031 “Effect of fixed orthodontic treatment on salivary nickel and chromium levels: A systematic review and meta-analysis of observational studies”

In this systematic review, the authors have investigated studies on nickel and chromium concentration in saliva as part of orthodontic therapy. The question is interesting, but there are some points that need to be clarified.

The introduction is too short and relatively unstructured. Please revise the introduction and try to insert a red thread into different parts of the Introduction.

Some parts of the discussion fit better into the Introduction.

Please also discuss the following point:

"in periods of two months and three months, salivary nickel level after treatment was less than that before treatment"

Author Response

Dear

1. We re-edited the English language.

2. We increase the introduction and revised it.

3. Several sentences of the discussion go into introduction. If introduction needs more description, please guide us.

4. We described “in periods of two months and three months, salivary nickel level after treatment was less than that before treatment” in discussion.

Reviewer 2 Report

I think the paper is quite thorough as is.  The authors described in detail how they identified the studies selected and analyzed the results in detail.  This was well executed and they detailed the studies with conflicting results, stating that more studies with larger sample sizes and longer followup times are indicated. The manuscript describes a thorough review of the literature analyzing salivary levels of nickel and chromium during fixed orthodontic treatment. The studies differ in timing, results and methods of analysis which is why there were only a few included in the final manuscript.

The strengths of the study are that there is a detailed analysis of the small number of  published data included. The authors describe  in great detail the results of the other authors.  The weakness is the varied results stemming from the small number of studies included.

Author Response

Dear

1. We re-edited the English language.

Reviewer 3 Report

Dear Authors,

the manuscript is interesting. Some questions raised.

Enlisted please find my considerations.

Introduction. Some studies evaluated metal release from orthodontic brackets, but in the Literature there is also Research about nickel and chromium release from orthodontic wires and ligatures. This concern could be added in introduction section.

Methods. Ok

Results. Ok

Discussion. Authors evaluated only in vivo studies. However, Many in vitro studies showed nickel (Prog Orthod. 2018 Feb 5;19(1):4. --- J Contemp Dent Pract. 2017 Mar 1;18(3):222-227. --- J Dent Res Dent Clin Dent Prospects. 2015 Summer;9(3):159-65. J Pharm Bioallied Sci. 2015 Aug;7(Suppl 2):S587-93. --- J Int Oral Health. 2015 Aug;7(8):14-20. --- Angle Orthod. 2014 Jan;84(1):140-8. --- Am J Orthod Dentofacial Orthop. 2010 Jun;137(6):809-15.  --- Angle Orthod. 2008 Mar;78(2):345-50.  ) and chromium (Prog Orthod. 2018 Feb 5;19(1):4. --- J Dent Res Dent Clin Dent Prospects. 2015 Summer;9(3):159-65. --- J Int Oral Health. 2015 Aug;7(8):14-20. --- Angle Orthod. 2009 Mar;79(2):361-7. -- Angle Orthod. 2008 Mar;78(2):345-50. ) release from orthodontic brackets. It could be useful to cite and discuss the results of the present report with in vitro results, in order to highlight similarities and differences.

Discussion. Some studies have tested nickel free orthodontic brackets, highlighting the differences with conventional brackets. This concern could be discussed in order to give a more complete view of the concern to the readers.

References. Some references could be added to strengthen discussion (some have been suggested above)

Figures. Ok

Tables. Ok.

Author Response

Dear

1. We re-edited the English language.

2. We described about wires and other orthodontics appliances in introduction.

3. We cited in vitro studies and a description about them.

4. Totally, we describe difference in release of ions from different brackets. If it needs more description, please guide us.                          

5. We added several new references.

Reviewer 4 Report

The work is interesting, but it is not sufficiently debated and explained why it was not found heterogeinity of the dosages of Ni and Cr in the periods: 10 min or less, 1 day, 3 weeks, 2 months and 6 months;  whyle large heterogeinity was found in the periods: 1 week, 1 month and 12 months.

Why does this happen?

What significance can it have from the clinical point of view and allergic sensitization?.

Author Response

Dear

1. We re-edited the English language.

2. We describe about difference about heterogeneities. If it needs more description, please guide us.